# Fluorescein Derivatives as Fluorescent Probes for pH Monitoring along Recent Biological Applications

**DOI:** 10.3390/ijms21239217

**Published:** 2020-12-03

**Authors:** Florent Le Guern, Vanessa Mussard, Anne Gaucher, Martin Rottman, Damien Prim

**Affiliations:** 1Institut Lavoisier de Versailles, CNRS, UVSQ, Université Paris-Saclay, 78035 Versailles, France; vanessa.mussard@universite-paris-saclay.fr (V.M.); anne.gaucher@uvsq.fr (A.G.); damien.prim@uvsq.fr (D.P.); 2Faculté de Médecine Simone Veil, Université de Versailles St Quentin, INSERM UMR U1173, 2 Avenue de la Source de la Bièvre, 78180 Montigny le Bretonneux, France; martin.rottman@aphp.fr; 3Hôpital Raymond Poincaré, AP-HP, GHU Paris Saclay, 104 Bd Poincaré, 92380 Garches, France

**Keywords:** fluorescein, pH-sensitive, probe, dyad, imaging, organelle, cell, bacteria, application

## Abstract

Potential of hydrogen (pH) is one of the most relevant parameters characterizing aqueous solutions. In biology, pH is intrinsically linked to cellular life since all metabolic pathways are implicated into ionic flows. In that way, determination of local pH offers a unique and major opportunity to increase our understanding of biological systems. Whereas the most common technique to obtain these data in analytical chemistry is to directly measure potential between two electrodes, in biological systems, this information has to be recovered in-situ without any physical interaction. Based on their non-invasive optical properties, fluorescent pH-sensitive probe are pertinent tools to develop. One of the most notorious pH-sensitive probes is fluorescein. In addition to excellent photophysical properties, this fluorophore presents a pH-sensitivity around neutral and physiologic domains. This review intends to shed new light on the recent use of fluorescein as pH-sensitive probes for biological applications, including targeted probes for specific imaging, flexible monitoring of bacterial growth, and biomedical applications.

## 1. Introduction

In chemistry, potential of hydrogen, or pH, is a data describing the acidity or the basicity of a medium [1]. Logarithmically obtained from H^+^ ion concentration, pH is one of the main physical characteristics used to describe an aqueous solution. Since the development of pH-meters, its measurement became inevitable across scientific fields, such as drinking water [2], industrial waste [3,4], global health [5], and agronomy [6]. As acidic or basic compounds are continuously released as outputs of cellular life, pH monitoring offered a unique opportunity to easily acquired data from biologic systems [7]. The relevance of pH in biological systems can be observed at different scales: the pH of biological fluids is well described and its regulation essential to the proper function of organs, since abnormal values are both the sign and cause of disease developments. pH regulation within biological systems relies on a sensitive equilibrium, called pH homeostasis. At the organism level, pH regulation is performed by the lungs through the elimination of CO_2_ and the kidney through the filtration of the HCO_3_^−^ ion. In cells, where organic acids, such as lactic, pyruvic, or beta-hydroxybutyrate acids are produced along metabolic pathways, some membrane proteins compensate the decrease of pH by transporting protons outside the cytosol [8]. Thus, cellular pH monitoring leads to the understanding of key milestones within cells, such as proliferation [9], ion transport [10], or carcinogenesis [11]. For example, unchecked biological processes occurring in malignant cells release a consequent amount of acid derivatives, leading to a pH decrease in tumoral tissues [12]. In microbiology, proliferations of aerobic bacteria also lead to a massive production of acidic metabolites, which quickly induce pH variations in medium [13]. Whereas global pH in cytosol has to be regulated, each organelle is fully effective in specific ionic environments, which are often correlated to local functions [14]. For example, the average pH in lysosomes, Golgi network, and mitochondria are 4.7, 6.7, and 8 respectively. Thus, pH monitoring of each organelle has helped to determine their function. At an atomic scale, all metabolism pathways are directly correlated to pH since enzymatic activities and kinetics depend on ionic environments. Another common example is the classification of essential amino acids, according their acidic or basic trends. Thus, pH is an important physicochemical datum, which can attest of biological activities at different scales. In order to monitor these fluctuations without any physical contact, fluorescent pH-sensitive probes were developed during the last century. Under specific light, these molecular probes re-emit photons at another wavelength for which related intensity depends on the surrounding pH [7]. Since optical devices are continuously becoming more precise, especially with the broad dissemination of optic fibers, pH-sensitive molecular probes are recurrent across biological studies.

One of the most used pH probes is fluorescein **1**; its chemical structure is composed of a tricyclic xanthene flanked by two hydroxyl groups and a bicyclic fused lactone fragment linked by a spiro carbon atom (Figure 1).

This family of molecules was discovered in 1871 by Adolf von Baeyer, and has become one of most ubiquitous probes in biological studies, because of its intense fluorescence, reversible pH sensitivity, chemical stability, and lack of cytotoxicity at working concentrations. For years, fluorescein **1** has been used as a starting material to create novel fluorescent probes revealing specific biological activities such as enzymatic cleavage [15]. Fluorescein **1** is still the focus of interest from the scientific community for its intense fluorescence and its sensitivity to pH variations around neutral domain [16]. As most biological systems are fully effective at physiological conditions (pH~7.3), fluorescein **1** became a benchmark for monitoring pH fluctuations in cell cultures. Indeed, every year, a substantial amount of new articles describes biological studies using fluorescein **1** as pH-sensitive probe (Figure 2) [17]. In this review, we propose an overview of fluorescein **1** and its main derivatives complemented by a survey of recent studies using fluorescein **1** as pH sensors for biological applications.

## 2. Fluorescein and Derivatives as Notorious pH Sensors

### 2.1. Fluorescein: Synthesis and Properties

Originally obtained by condensing phtalic anhydride with phenol in acidic conditions by Von Baeyer, this preparation of fluorescein **1** is nowadays based on Friedel–Crafts reactions (Figure 3). Thus, the fluorophore can be easily obtained by mixing phtalic anhydride, resorcinol and ZnCl_2_ or methanesulfonic acid [18,19]. Large scale manufactured, fluorescein **1** has been included in the World Health Organization (WHO)’s model list of essential medicines [20] for its biological applications, but it is also used in other fields, such as petrochemistry for leak detection and cosmetic formulations.

Under basic conditions (pH > 8), fluorescein **1** absorbs blue light with a maxima absorption peak around 490 nm, and emits a green light around 515 nm [21]. Due to phenolic fragments and equilibrium between a carboxylic function and a lactone, ionic charge and chemical structure of fluorescein evolve depending on the surrounding pH, leading to fluctuations of photophysical properties (Figure 4) [22]. Whereas the fluorescence quantum yield under an excitation at 490 nm is very high under basic conditions (Φ_F_: 0.95 in NaOH 0.1 M), an acidification of the solution progressively leads to the fluorescence extinction [23]. This fluctuation is due to the transition of the di-anionic form of fluorescein into an anionic equilibrium, which has a lower absorbance associated to a blue-shifting [23].

The most problematic drawback of fluorescein is its photobleaching when exposed to light [24]. Indeed, the fluorescent emission from this probe progressively decreases under intense light irradiations, which is an issue since pH monitoring is based on this intensity, and repeated measurements cause the signal to fade. Because of this photobleaching, all fluorescein derivatives must be stored in the dark, and experiments using this probe have to occur quickly. Spectral bands of fluorescein can also induce issues with some optical systems. Depending on the purpose of the application, its relatively broad fluorescent emission can be considered as an advantage (for example, in case of the Förster resonance energy transfer (FRET) system), but also as a drawback for multi fluorescent probing experiences. Fluorescein presents also a weak Stokes shift, the interval between highest absorption and emission wavelength, which is a potential issue for devices using a single optical path, such as entry-level plate readers. Like most intense fluorescent probes, self-quenching can occur in case of aggregation (ACQ) or high degrees of surface substitution [25]. Hopefully, due to low the concentration used in biological systems, this drawback has little relevance.

### 2.2. Most Used Fluorescein Derivatives and Their Properties

Several fluorescein derivatives have been developed in order to fit with the downstream applications. The best-known derivatives are based on the modification of benzo-fused lactone moiety in order to add reactive chemical functions such as isothiocyanate, carboxylic acid, or amine. The outcoming probes are named fluorescein isothiocyanate (FITC) (Figure 5-**2**), carboxyfluorescein (5(6)-FAM or CF) (Figure 5-**3**), and fluoresceinamine (FA) (Figure 5-**4**), respectively.

FITC **2** is one of most used derivatives due to its properties and good reactivity for conjugation. In comparison to fluorescein, FITC **2** presents a slight decrease of fluorescence quantum yield (Φ_F_ (excitation: 500 nm): 0.75 in 10 mM Phosphate-Buffered Saline) [26]. However, in comparison with other fluorophores, FITC **2** still has an intense fluorescence and can be used as a pH-sensitive probe around the neutral domain. FITC **2** can be easily associated to any kind of chemical structures bearing an amine fragment, such as fluorophores [27], targeting agents [28], proteins [29], polymers [30], or nanoparticles [31]. Resulting bioconjugation linkage is based on a thiourea function for which stability depends on physiochemical surroundings due to its strong H-bond interactions [32,33,34]. Using such derivatives offers a unique opportunity to create specific pH sensors, and particularly, for biological systems where all proteins are bearing a terminal amine. Even if FITC **2** is still the most prolific derivative in literature (Figure 6), new fluorescein reactive derivatives are still described with optimized properties [35].

When fluorescein was used for intracellular pH (pH_in_) monitoring, a high leakage rate appeared from the cells, which made accurate pH determination difficult. Fluorescein **1** was then replaced by carboxyfluorescein **3** (FAM or CF) as the main pH_in_ probe, since its main advantage was its low leaking rate through cell membranes. Indeed, due to its additional carboxylic acid function, the supplementary anionic charge significantly reduces solubility of FAM **3** in lipid structures composing cell membranes [36] Moreover, FAM **3** has similar photophysical properties to FITC **2** [26]. Thus, FAM **3** has become one of the most used probes for characterization of tumoral tissue [37]. FAM **3** can also be used for preparation of bioconjugates using carbodiimide/NHS activation. In this case, resulting amide function is stable due to its omnipresence in biological systems. Diacetate derivatives of fluorescein and FAM **3**, respectively named fluorescein di-acetate (FDA) and carboxyfluorescein di-acetate (CFDA) (Figure 7-**5**), were also been used across several studies [38,39]. Di-ester analogs present a quenching of the fluorescence, but, under esterase intracellular activities, emissions are restored [40]. Thus, these derivatives are used to estimate intracellular enzymatic activities [41]. Moreover, in comparison of FAM **3**, which is used for its weak membrane permeability, diacetate derivatives are cell permeable probes. Thus, CFDA **5** is one of best pH_in_ sensors because FAM **3** is directly released under intracellular enzymatic activities and is not able to leak outside cells.

While FITC **2** and FAM **3** are the most prominent fluorescein derivatives in the literature, other probes were developed for specific and optimized applications at physiological pH. Introduced by Roger Tsien in 1982, 2′,7′-bis-(2-carboxyethyl)-carboxyfluorescein (BCECF) (Figure 7-**6**) was developed in order to fit more precisely with pH variation around neutral domain [42]. Indeed, due to its pKa near 7, weak acidification of medium directly induces a significant reduction of fluorescence intensity. Moreover, the absorption spectrum of BCECF **6** possesses an isosbestic point, where its absorbance is pH-independent. This property allows the establishment of ratiometric analyses for which a ratio of fluorescence is calculated from excitations at two different wavelengths (generally at maxima and isosbestic point). These protocols greatly overcome common hurdles that are dye loading, leakage, optical imprecisions, or photobleaching [43].

In order to fit with acidic organelles, present in yeast [44] for example, fluorinated derivatives of fluorescein, such as Oregon Green **7**, have been prepared for acidic pH monitoring (Figure 7-**7**) [45]. Introduction of electron-withdrawing atoms within the xanthene structure leads to a decrease of pKa around 4.7. Then, pH-sensitivity of such compounds is suitable with cellular components operating at pH around 5.

## 3. Recent Studies Using Fluorescein as pH Sensors for Biological Applications

### 3.1. Cellular pH Imaging

#### 3.1.1. Molecular pH Sensor

Most of recent developed pH-sensitive probes are based on ratiometric determination. This strategy spread across biological studies because previous developments using simple fluorescence monitoring were subject to several interferences such as the quality of the optical devices used, the biological and physical environments, and compounds stability (such as photobleaching or chemical degradation). To overcome these issues and increase probe precision, ratiometric measurements have been successfully adopted. The most convenient way to use this method is to associate a pH-dependent probe to one (dyad) or more fluorophores emitting at different wavelengths under the same excitation. The ratio of fluorescence is then calculated in which pH-independent probes are used as internal control.

Novel chemical associations based on fluorescein **1** were recently elaborated as dyads. For example, Zhang et al. proposed a ratiometric probe based on the chemical association **8** of an iridium (III) complex with FITC **2** (Figure 8) for pH_in_ monitoring [27]. Under a single wavelength excitation at 488 nm, ratiometric determination of pH_in_ is possible from 5.2 to 7.8, using near infrared emission from iridium (III) complex.

In addition to the creation of ratiometric probes, the dyads strategy can also be an opportunity to enlarge operating windows of pH-sensitivity of fluorescein. Since large pH fluctuations may occur in cells, during mitophagy for example, more accurate and wider pH-sensitive monitoring offers a better understanding of cellular life. However, the range of the pH efficient interval for fluorescein **1** is limited to 6 to 8. Therefore, Lee et al. put forward a novel dyad based on chemical association with rhodamine B (**9**) (Figure 9) [46]. Inspired by previous works using both compounds grafted on nanodots [47], rhodamine was covalently linked to fluorescein **1** in order to create a pH-sensitive probe efficient from 4 to 8. The widening of the pH efficient interval is based on rhodamine fluorescence, which decreases with alkalization. Moreover, both fluorophores have similar absorbance properties, offering an opportunity to use ratiometric calculation by exciting both molecules at 488 nm. Finally, the dyad was shown to be effective to determine pH_in_ across in vitro experiments.

Another approach was proposed by Wu et al. to widen the interval of pH sensitivity [48]. Two pyrene groups were covalently linked to FITC **2** (**10**) in order to create a time-resolved FRET system (Figure 10). Under an excitation at 358 nm, bispyrene moieties have an emission peak at 459 nm fitting with FITC **2** absorbance. Due to bispyrene lifetime properties, excitation of FITC **2** by FRET occurs following alkalization of the solution. Thus, ratiometric determination based on both emissions is effective for pH from 3 to 10. Cellular experiments showed that this probe can be well adapted to pH_in_ determination between 4 and 8.

Mitochondrion is a prominent organelle for eukaryotes organisms because of its major role in respiration systems, energy production, enzymatic activity, and cations storage. Monitoring pH fluctuations within mitochondria is an important challenge considering the importance of the chemical processes involved. Previously, Carboxy-SNARF^®^ (Seminaphtharhodafluor) was the preferred mitochondrial pH-monitoring probe due to its passive accumulation [49], but was deemed insufficiently selective or rapid for this specific application. In order to improve mitochondrial pH monitoring, different approaches were recently proposed. Chen et al. described a dyad based on the chemical association of cyanine with FITC **2**, called “Mito-pH” (Figure 11-**11**) [50]. In addition to its ability to selectively target mitochondria [51], cyanine has pH-insensitive fluorescent properties allowing a ratiometric monitoring. The ratiometric signal was described as a linear response from pH 6.1 to 8.4 across in vitro experiments. Yet, Li et al. described a ratiometric probe by broadening fluorescein absorbance with chemical coupling of unsaturated indolium (Figure 11-**12**) [52]. As previously, such structure tends to accumulate within mitochondria due to its lipophilic cationic structure [51]. This probe showed a repetitive signal even in highly ionic solutions. After full characterization of the ratiometric signal for pH varying from 4 to 10, confocal imaging showed the capacity of this probe to selectively reach its target. Chloroquine treatments were applied to cells in order to simulate pH fluctuations, since this compound induces intracellular alkalization [53]. This novel probe has proved its ability to track in real-time wide mitochondrial pH fluctuations. In order to increase the Stokes shift, Qi et al. proposed the hybridization of the fluorescein with a coumarin moiety (Figure 11-**13**) [54]. Indeed, overlapping of emission and excitation spectra, as in case of fluorescein, may induce the requirement for high-resolution optical devices. Thanks to its hybridization between both fluorophores, the resulting probe was effective in the determination of mitochondrial pH, with maxima wavelengths about 450 and 550 nm for absorption and emission respectively.

#### 3.1.2. Supported Sensors

For years now, nanoparticles attracted a lot of attention because of their applications in cancer therapy [55,56]. In addition to physical and chemical advantages inducing flexible and graftable surfaces, their biocompatibility allows them to massively penetrate tumor cells. Thus, multifunctional nanoparticles were quickly described for therapeutic and diagnostic applications. Therefore, novel pH-sensitive nanoparticles bearing fluorescein moieties are continuously described. Because pH measurement within lysosomes is essential to understand cellular metabolisms, particularly in tumoral cells, Zhang et al. designed novel pH-sensors based on silicon nanodots bearing aptamers and FAM **3** (**14**) (Figure 12) [57]. The use of the aptamer (AS1411) enhances cellular uptake and specifically targets tumors and lysosomes due to its strong affinity for nucleolins. Because of native fluorescence properties from these particles, only grafting of FAM **3** was required to create a ratiometric system. The preparation of final nanodots was easily established using simple reactions such as formation of an amide and thiol–Michael click chemistry. Using ratiometric calculations between emissions from nanodots and FAM **3**, estimations of lysosomal pH were performed. Since aptamers grafted on dots have the ability to recognize malignant cells, an observable difference of fluorescence between the imaging of human breast MCF-7 cancer cells and normal cells MCF10-A was easily observed. Thus, such particles offer an interesting way to selectively image cancer cells and acquire large amounts of data concerning their lysosomal in-situ pH variations.

Whereas aptamers are used as targeting agents in this previous study, Ding et al. proposed their own ratiometric pH-sensitive probes specific to cancer cells using folic acid [58]. Due to high density of folate acceptors in cancer cells, gold nanoclusters (AuNCs) covered by bovine serum albumin (BSA), FITC **2**, and folic acid, were designed to selectively penetrate their target, providing a ratiometric determination of pH_in_. Once again, native properties of the used nanoparticles fit perfectly with FITC **2**, since a single excitation at 488 nm was enough to induce fluorescence from the clusters and the pH-sensitive probe. Efficient for pH from 6.0 to 7.8, this system was tested over HeLa cells for which pH variations were simulated using ammonium chloride treatments.

Whereas most studies are focused on intracellular data, Yang et al. proposed a ratiometric probe called “FITC-FPen/FPen@AuNC” for determination of extracellular pH (pH_ex_) [59]. Ionic concentrations surrounding cell surface is also a parameter, which can indicate metabolic disorders such as tumor metastasis [60], ion transits deregulation [61], or even some virus infections [62]. A similar chemical approach to the study from Ding [58] was proposed here; BSA protected AuNCs, were covered by FITC and cell penetrating peptides (CPP). CPP are cationic peptides, which are generally used as carriers for transporting compounds through membranes. In this case, the used peptide (FPen) is ensuring agglomeration of nanosensors at the cell surface whereas emissions from AuNCs and FITC **2** are providing the ratiometric determination of pH_ex_. Confocal imaging of HeLa cells exposed to different media proved the good sensitivity and low cytotoxicity of theses nanoparticles (Figure 13). Another complementary strategy was proposed by Ohgaki et al. using FITC-poly(ethylene glycol)-phospholipid derivatives [63]. This small and amphiphilic polymer has the ability to be inserted into plasma membrane of cells, and then grafted FITC can be used as probe for determination of pH_ex_. The cell-surface labeling with this polymer presented many advantages such as its sensitivity, pH-reversibility, and its generic application.

Immobilization of fluorescein for recurrent surface pH determination displays many challenges because it should avoid leaching, degradation, or photobleaching. Preparation of fluorescent pH-sensitive surfaces is also offering additional benefits since it can be used as optic fiber coating or in bioanalytical protocols, such as evanescent wave sensors. Bidmanova et al. investigated a general procedure for immobilization of fluorescein derivatives on surfaces, by using BSA derivatives [64]. Thus, FAM-BSA conjugates were immobilized on glass surface using direct glutaraldehyde cross-linking or after incorporation into ORganically MOdified CERAmics (ORMOCER) (Figure 14). The resulting surfaces have proved to be highly suitable for their mechanical stability, negligible (photo)leaching, and fluorescent pH-sensitivity from 4.0 to 9.0. Due to the ease of use, such protocols may lead to the preparation of a supported sensor for a wide range of applications including bioprocessing, biochemical analyses, environmental analysis, or healthcare devices.

### 3.2. Dual Sensors

The development of ratiometric probes is related to the recent improvement of optical devices composing imaging systems. Following this increasing precision, more studies propose to add other sensitive probes to fluorescein-based compounds. Thus, such dual sensor offers the benefit of two physiochemical data in a single optical read, which is a unique opportunity for the understanding of biological systems.

Using AuNCs, Han et al. described the preparation and characterization of a ratiometric probe **15** providing intracellular dual determinations of Cu^2+^ concentration and pH [65]. Three different molecules were grafted on these nanoclusters, firstly FITC **2** as pH-sensitive probe, secondly a tailor-made specific Cu-ligand (called TPAASH) and thirdly a coumarin derivative as a reference probe (Figure 15). Monitoring of Cu^2+^ cations is based on the quenching of AuNC emission at 722 nm (under excitation at 405 nm), following the increase of copper concentration. Therefore, coumarin was also grafted on clusters to be used as reference probe since its excitation wavelength is also about 405 nm and emission is at 472 nm. These novel particles allow the ratiometric determinations of pH_in_ from 6 to 9, and simultaneously, of Cu^2+^ concentration up to 11 µM. A few years later, Zhu et al. proposed their own version of a dual pH/Cu^2+^ sensor by grafting FITC **2** and polyethylenimine (PEI) on carbon dots [66]. Due to a ratiometric and linear signal, they were able to determine pH and Cu^2+^ concentration into yogurt and human serum samples.

Oxygen is a key component for biological systems, and especially, for mammal cells for which metabolism pathways are based on Krebs cycle. Thus, dual sensing probes allowing the simultaneous monitoring of oxygen concentration and pH at the same time are very important in biomedical sciences. Relevant publications reported the creation of such dual sensors [67,68,69] based on the combination of fluorescein with O_2_-sensitive fluorophores, such as platinum and ruthenium complexes. In order to improve dual monitoring of pH_in_ and O_2_, Xu et al. described the use of semiconducting polymer dots **16** (Pdots) as a support of both sensitive probes, since these nanomaterials have an improved cell uptake [70]. For this system, FITC **2** and a platinum porphyrin complex were targeted as pH and O_2_ sensors respectively (Figure 16C). Once again, the native photophysical properties of the dots were employed; based on a FRET effect, this multimodal sensor allows ratiometric determination under excitation at 405 nm, since the fluorescence of Pdots (410 to 470 nm) is matching the absorption of FITC (410 to 500 nm). Thus, one single excitation at 405 nm leads to emissions from all sensors at different wavelengths (blue for Pdots, green for FITC **2**, red for porphyrin). After calibration, in vitro experiments on CaSki cells proved that Pdots are well embedded in the cells, allowing the improved monitoring of both physical parameters using ratiometric results (Figure 16A,B).

A promising approach, described by Meier et al., focuses on imaging devices in order to obtain signals from dual sensors using a conventional digital camera [71]. A sensor film was created by the incorporation of microparticles bearing sensors: platinum porphyrin complex for oxygen sensing, FITC **2** for pH sensing and diphenylanthracene as reference. Under light irradiation at 405 nm, emissions in red green blue (RGB) channels can be recorded by a digital camera. After characterizations, calculations and calibrations, the proposed system was even used across in vivo experiments (Figure 17). The dual sensor film was applied on healing wounds where inflamed tissues were directly submitted to abnormal pH and oxygen concentrations. Furthermore, the same team developed a diligent expertise using such systems, by proposing various optimizations, such as a sprayable version [72], or applications concerning chronic wounds [73], water plants [74], and recently, radiation therapy [75].

In case of cancer cells, heat production is higher than in healthy cells due to abnormal and aroused metabolisms [76]. Thus, the developments of dual sensors monitoring pH and temperature have also been investigated. In this context, Rhodamine B is one of the most notorious temperature-sensitive fluorescent probes. Liu et al. described the preparation of polystyrene microbeads embedded with rhodamine, which were subsequently coated with FITC **2**. This system presents a linear fluorescence response from rhodamine and temperature from 32 to 38 °C. The physicochemical stability and the selective response from each probe are among advantages of such combination. Later, Zhang et al. proposed their own dual sensitive system using similar probes, but of a reduced size [77]. In addition to the beneficial nanoscale of this sensor, a europium complex was included as a reference dye in order to offer a ratiometric determination. Moreover, the surface of this nanosensor was coated with cationic charges in order to increase its affinity for lysosomes. Across a series of in vitro experiments using HeLa cells, his dual sensor was able to determine pH ranging from 4.0 to 9.0, and temperature between 25 and 40 °C.

### 3.3. Bacterial Growth

Monitoring aqueous physicochemical data, such as pH, is also a significant opportunity to understand bacterial growth. Indeed, due to their capacity to transform and adapt their environment through their quick proliferations and the production of extracellular compounds, ionic variations cause pH changes in a few hours [78]. The modification of pH along incubation periods has been widely used to monitor microorganism growths in various media, including in blood cultures. In fact, it is usually used to detect the presence of bacterial contamination in medical [79], food [80], and water [81] sectors, which is becoming more of a relevant issue because of emergence of bacterial resistance [82]. Thus, pH-sensitivity around the neutral domain of fluorescein perfectly fits with microbial culture, since most of in vitro experiments using common strains are conducted at physiological pH. For example, Si et al. designed nanoparticles bearing fluoresceinamine (FANPs) for an accurate real-time detection of *Escherichia coli* growth (Figure 18) [83]. Resultant non-toxic and fluorescent polystyrene nanoparticles, obtained after grafting FA **4** by carbodiimide coupling, were incorporated in in vitro cultures, where emitting fluorescence was directly linked to medium pH. Following the proliferation of *Escherichia coli*, the pH is decreasing which directly reduce the fluorescence of the particles. Thus, the simple fluorescence from culture suspensions is sufficient to determine if there is any growth or inhibition of the strain. This system was also challenged using cultures exposed to different concentrations of antibiotics, where pH (and fluorescence) kinetics were precise enough to determine bacterial behaviors. Thus, such particles can easily be used for the screening of potential growth inhibitors or for the determination of bacterial minimal inhibitory concentration.

In their view, Wang et al. focused on pH monitoring in case of bacterial growth on Petri dishes cultures [84]. Since their creation in 1887, Petri dishes are still the most widely used support for in vitro cultures due to their flexibility fitting the broad microbial diversity. In addition to agarose and appropriate nutriments, Wang et al. proposed to add silicone nanoparticles bearing FA **4** and porphyrin moieties in the preparation. These nanosensors are physicochemical stable, easy to incorporate within agar preparation, are not internalized by bacteria, and are based on ratiometric method. Thus, this study offers an effective protocol to design pH-sensitive Petri dishes, which can be used to visualize bacterial growth or determine resultant pH variations due to bacterial behaviors (Figure 19).

Some strains have the ability to adhere on surfaces and form a biofilm throughout the production of an extracellular matrix promoting the survival, progression and growth synchronization of the colony. Bacterial biofilms are a major medical concern because of their ability to colonize indwelling devices, most importantly catheters and implants [85]. In order to study the metabolisms and mechanisms implicated in such organized structures, in vitro biofilm cultures have showed their utility, as well as the development of suitable fluorescent probes. Gashti et al. described the preparation of a pH-monitoring microfluidic platform adapted for biofilm studies [86]. To reach this goal, FITC **2** based silver core-silica shell nanoparticles were prepared, and then, covalently linked to glass substrates using click chemistry. This microfluidic system has many advantages, such as physicochemical stability, lack of leakage of/from nanoparticles (NPs) and real-time pH measurement. For example, dynamic pH responses from an oral biofilm of *Streptococcus salivarius* were monitored following exposition to different glucose concentrations (Figure 20). Offering bacterial biofilm monitoring, such described system has also advantages linked to microfluidic scales where liquids consummations are reduced, cultures improved due to laminar flows and reaction kinetics optimized [87,88]. Moreover, this study is also suggesting that, due to the use of nanoparticles as platform for fluorescent probes, additional sensitive probes may be also integrated to create a multi-modal microfluidic system and to increase income of information.

In comparison with previous reported studies in this review, targeted or optimized probes are rarely developed for imaging specific bacterial organelles. Due to their small size and their potential mobility during imaging experimentations, bacteria do not fit perfectly with specific organelle monitoring. Furthermore, diacetate derivatives (FDA and CFDA **5**) are perfect candidates for monitoring bacterial enzymatic activity, explaining why recent studies are mainly using fluorescein at its purpose [89,90,91,92,93]. However, some work also used these fluorescein derivatives for their pH-sensitivity.

A wide diversity of bacterial species exists across all biological systems. Each local microbiome is directly linked to the properties of its surroundings, since only bacteria supporting local physicochemical properties are able to fully grow. Thus, using pH-sensitivity of fluorescein along in vivo assays can be used as a way to determine the suitable environment for each strain. For example, aciduric bacteria are generally involved in buccal microbiome, as described in the study below. Among buccal strains, *Streptococci mutans*, *Bifidobacterium dentium*, and *B. longum* have been observed in case of caries lesions, often linked to local acidification. In order to compare their survival rate in acidic environment, Nakajo et al. proposed protocols using CFDA **5** [94]. Using pH sensitivity of fluorescein, the difference between pH_in_ and pH_ex_ offers a way to observe bacterial ability to protect itself against acidification. In this case, it appeared that *Bifidobacterium* strains are the most stable species in acidic environment explaining why they are predominant in case of caries lesions.

Using similar protocols based on comparison between intra and extracellular pH, bacterial life cycle can be also monitored since substrate consummation, and resulting productions of metabolites may induce variations in both environments. For example, Bouix et al. described a protocol to monitor pH_in_ variations following the malolactic fermentation (MLF) of *Oenococcus oeni* in wine [95]. Using diacetate derivatives of carboxyfluorescein **3** and Oregon Green **7**, pH_in_ variations from 3 to 6 were accurately observed using flow cytometry analysis along different growth phases. It appeared that fermentation occurs during the exponential phase, where malic acid is co-excreted as lactic acid, carbon dioxide, and proton. This production induces an increase of pH_in_ and stimulates the ATP synthase pathway. This study shows that pH monitoring using fluorescein derivatives provides valuable insights into the cellular mechanisms at play in pathogenesis.

### 3.4. Other Applications

Following the principle of drug vectorization, any component or organelle from biological systems may be targeted by fluorescein derivatives in order to monitor its pH variations. For example, Li et al. recently described a novel pH-sensitive probe targeting bones [28]. It appears that pH is a relevant datum concerning bones homeostasis [96] or abnormal issues, such as metastasis [97]. To create an efficient bone pH-sensitive probe, FITC **2** was conjugated to alendronate, a bisphosphonate compound presenting a significant affinity for hydroxyapatite (HAp) (**17**) (Figure 21). The resulting probe was able to selectivity target bones, without interfering with other calcium compounds or organs. Nude mouse models were tested in vivo and pH variations from 6.8 to 7.4 were observable. The use of a ratiometric version would provide a powerful diagnostic tool, and this probe is a remarkable example of fluorescein vectorization for pH monitoring.

The design of fluorescein based ratiometric sensors was applied to the development of novel imaging tools. Mathew et al. described a novel nanoprobe for imaging and monitoring pH variations within *Caenorhabditis elegans*, a nematode usually used as in vitro model organism [98]. Monitoring intestinal intracellular pH of natural strains of *C. elegans* was quite challenging, because of the presence of a protecting and surrounding cuticle, and selective intestinal uptake [99]. To overcome these issues, nanocolloidal silica particles bearing FITC **2** and rhodamine patterns were prepared. Thanks to the intrinsic properties of the nanoparticles, the fluorophores are physicochemically stable and able to bypass the selective intestinal barrier. The resulting probe was successfully challenged by using two different mutants (*eat-3* and *N2*), considering that first strain presents a lower pH due to fragmented mitochondria. Thus, this study proposes a new and non-invasive way to monitor the pH_in_ of *C. elegans*, which would interest all further studies using this specie as model organism [100].

Besides the medicinal applications described above, pH monitoring is also a useful tool in odontology. The dental microbiome is a symbiotic partner with our organism [101]. However, the proliferation of aciduric bacteria within a sturdy biofilm can metabolize sugars into organic acids, which may lead to demineralization of enamel [102]. In order to prevent any stubborn damage to the dentition, such as caries, the healthcare specialists can be provided with a tool to evaluate the production of organic acids from dental microbiome under sugary environment. Furthermore, conventional procedures, using pH paper for example, are not readily applicable to the dental topography. Thus, Sharma et al. worked on an optical system involving fluorescein **1** in order to monitoring dental pH [103]. Fluorescein **1** fits perfectly the requirements of this use case thanks to its regulatory approval (FDA, EMA) for internal use in humans. The main attention was focused on the creation of the fiber optic-based dental probe and post data processing. Based on a ratiometric calculation using anionic and dianionic forms of fluorescein, this device allowed the monitoring of dental pH across pilot studies, whereas acidic production was observable following sucrose rinse (Figure 22).

Obviously, the simplest use of fluorescein pH-sensitivity should be acidic or basic titrations around neutral domain. Whereas colorimetric titrations in biological medium require further calibrations due to irregular native absorptions, fluorescent titrations are convenient to use. An example of titration was described by Burton et al. in order to determine sarcosine concentration in urine [104]. Previously debated, sarcosine is nowadays recognized as an accurate marker of development of prostate cancers [105,106,107]. In order to reduce the cost of conventional techniques based on chromatography, fluorescent titration was established using enzymatic transformation of sarcosine in formaldehyde, which is used as a reactant for the production of formic acid. Therefore, fluorescent intensity from fluorescein is directly proportional to initial amount of sarcosine (Figure 23). Despite an expected lower sensitivity of this protocol in comparison to chromatographic systems, associating enzymatic transformation with fluorescence titration should be considered for its efficacy, efficiency, practical, and flexible procedures.

While BCEG **6** is one of most sensitive pH fluorescent probes around the neutral domain, a new fluorescein-like derivative called SNARF^®^ (**18**, Figure 24) represents a recent breakthrough. With a pKa similar to BCEG, this probe offers a double fluorescent emission signal at 580 and 640 nm for one single excitation at 488nm. This double signal allows the estimation of pH_in_ using a ratiometric calculation without any supplementary probe. For example, Golda-VanEeckhoutte et al. recently showed that SNARF^®^
**18** was more efficient for the determination of pH_in_ in phytoplankton, than BCECF **6** [108]. By using an acetate ester analog, the pH sensor is accumulating inside cells under the activity of esterase. The intracellular probe concentration is based on enzymatic kinetics, leading to fluorescence fluctuations independently from pH variations. The intrinsic ratiometric property from SNARF^®^ overcomes this issue. Thus, this derivative offers an easy and reliable solution for monitoring of pH_in_ in phytoplankton, which allows observation of cellular metabolic processes responding to the variation in oceans.

## 4. Conclusions

This review is intended to provide an overview of recent advances in the field of fluorescein derivatives, their fluorescent pH-sensitive properties, and applications in biological systems. Despite its discovery in 1871, fluorescein and its derivatives attracted lasting interest from the scientific community over the past decades. The development of new fluorescein derivatives remains relevant since the properties of members of this family of probes perfectly fit with changes in pH around the neutral and physiological domains. New fluorescein-based dyads for the creation of optimized or targeted probes have been one of main recent focus. Nanoparticles bearing fluorescein moieties, increasing its stability and offering a broad spectrum of applications, represents one of the most recent advances in this field. Chemical combinations or grafting mostly used FITC or FAM derivatives, but some novel fluorescein derivatives have been described in order to modulate photophysical properties, improve measurements accuracies, and expand the fields of application. Results and findings gathered within this review undoubtedly suggest that fluorescein will continue to be an essential pH-sensitive probe in the future.

## Figures and Tables

**Figure 1 ijms-21-09217-f001:**
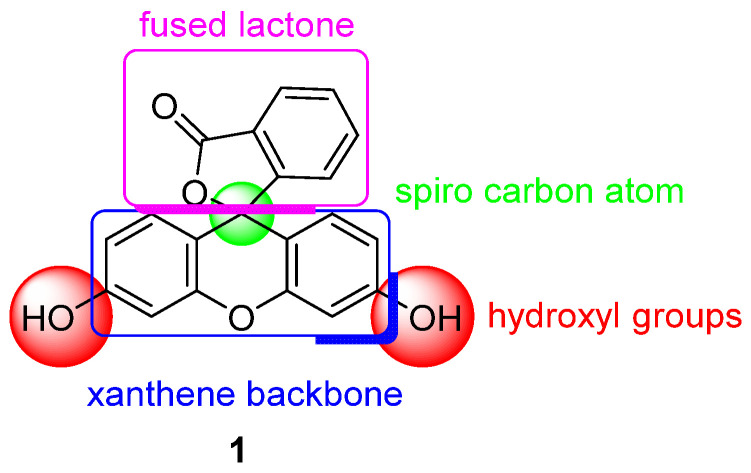
Chemical structure of fluorescein **1.**

**Figure 2 ijms-21-09217-f002:**
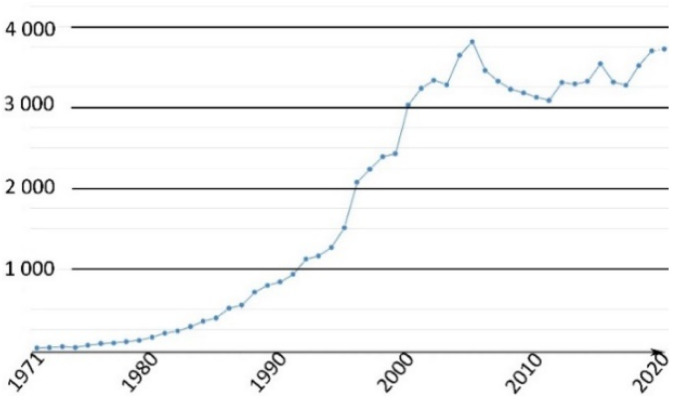
Number of articles with keywords “fluorescein, pH, probe, biology” during last fifty years [17].

**Figure 3 ijms-21-09217-f003:**
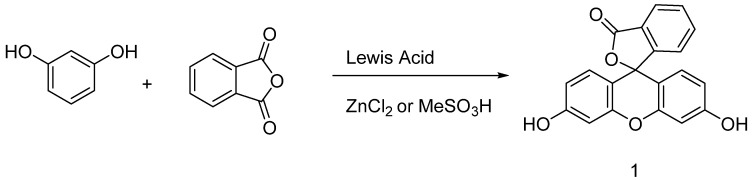
General synthetic pathway to fluorescein.

**Figure 4 ijms-21-09217-f004:**
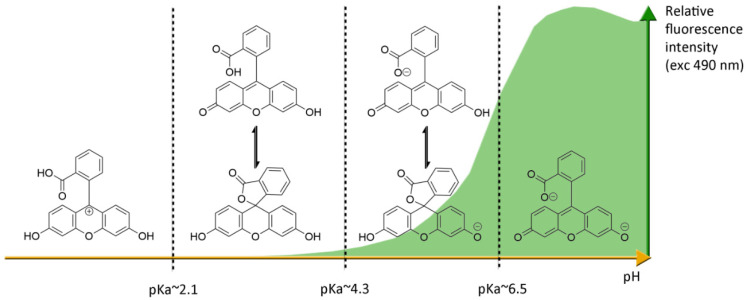
Ionic forms of fluorescein **1** according the pH domains and their relative fluorescence intensities. At neutral pH and under excitation at 490 nm, the most fluorescent di-anionic form of fluorescein takes prominence over other forms. Below pH = pKa~6.4, mono-anionic fluorescein displays a blue-shifted absorption followed by drastic decrease of fluorescence. At even lower pH, neutral and further cationic forms of fluorescein becomes non-fluorescent under irradiation at 490 nm.

**Figure 5 ijms-21-09217-f005:**
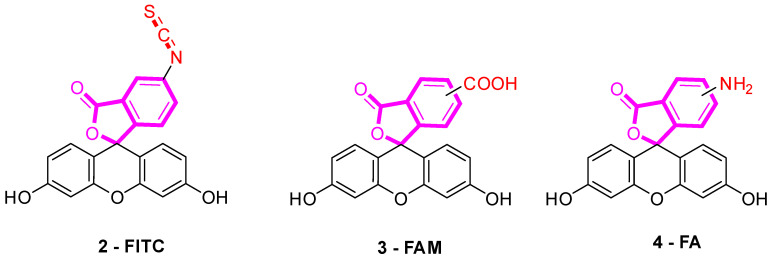
Most used reactive derivatives of fluorescein.

**Figure 6 ijms-21-09217-f006:**
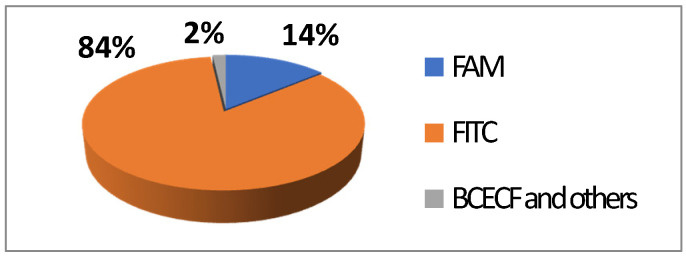
Average repartition of used fluorescein derivatives in literature the last decade [17].

**Figure 7 ijms-21-09217-f007:**
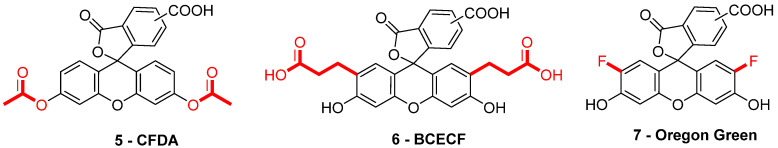
Different fluorescein derivatives used as pH-sensitive probes.

**Figure 8 ijms-21-09217-f008:**
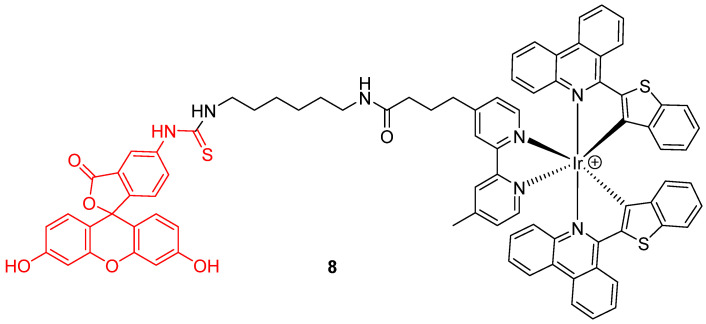
Iridium (III) complex-fluorescein dyad **8** as intracellular pH (pH_in_) probe by Zhang [27].

**Figure 9 ijms-21-09217-f009:**
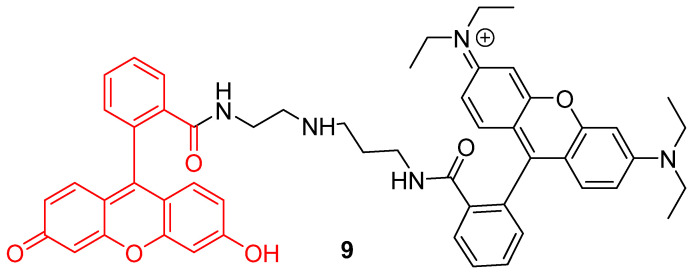
Chemical structure of rhodamine-fluorescein dyad **9** from Lee [46].

**Figure 10 ijms-21-09217-f010:**
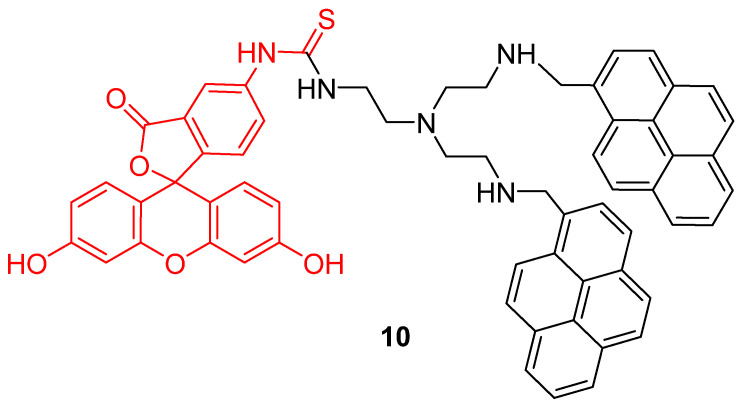
Bispyrene-Fluorescein dyad **10**, as Förster resonance energy transfer (FRET) based pH-sensitive probes by Wu [48].

**Figure 11 ijms-21-09217-f011:**
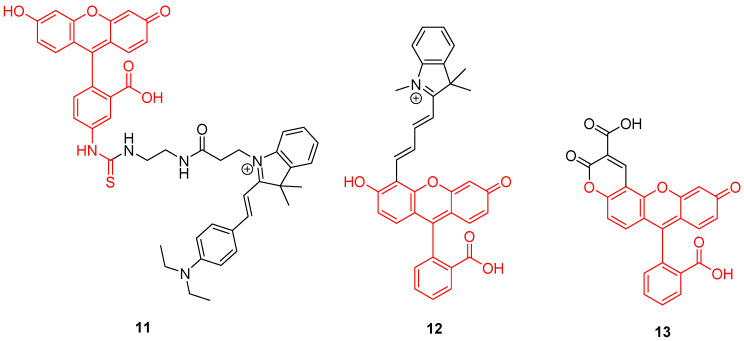
Mitochondrial pH-sensitive probes according to Chen [50], Li [52], and Qi [54].

**Figure 12 ijms-21-09217-f012:**
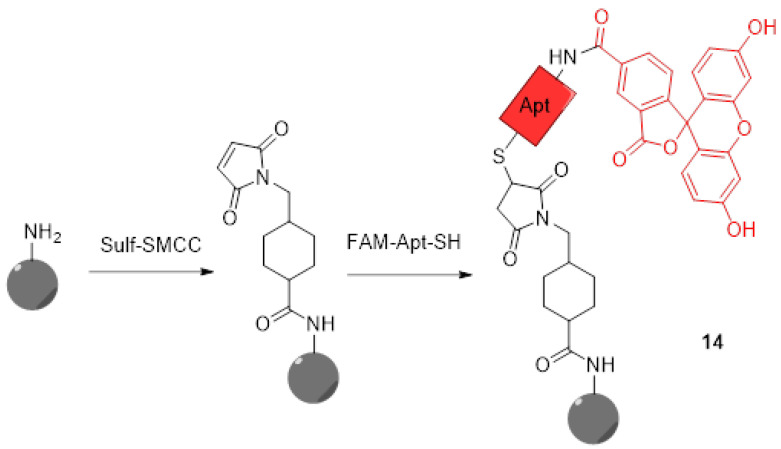
Preparation of silicon nanodots **14** bearing aptamers and FITC according to Zhang [57].

**Figure 13 ijms-21-09217-f013:**
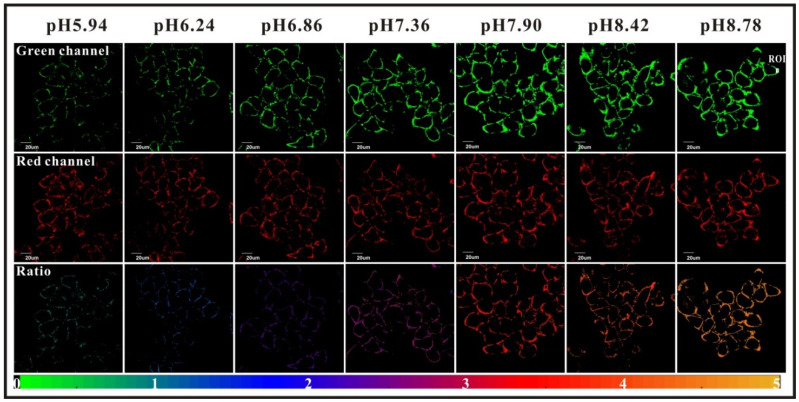
HeLa extracellular pH determination using FITC-FPen/FPen@AuNC. Reproduced with authorization from Yang [59].

**Figure 14 ijms-21-09217-f014:**
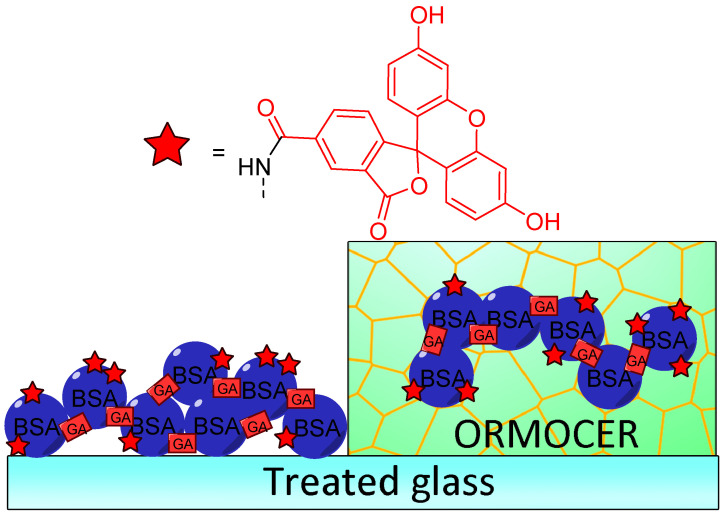
Immobilization of carboxyfluorescein (FAM)-bovine serum albumin (BSA) on treated glass by direct glutaraldehyde (GA) cross-linking or after incorporation in ORganically MOdified CERAmics (ORMOCER), according to Bidmanova [64].

**Figure 15 ijms-21-09217-f015:**
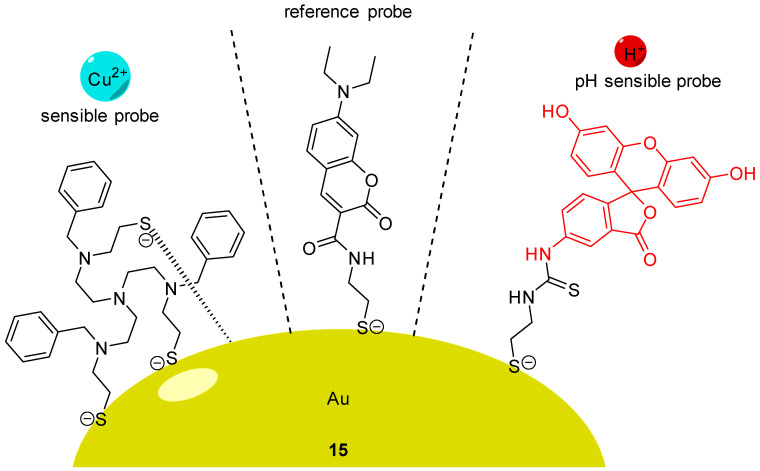
pH and Cu^2+^ sensitive nanoprobe **15** according to Han [65].

**Figure 16 ijms-21-09217-f016:**
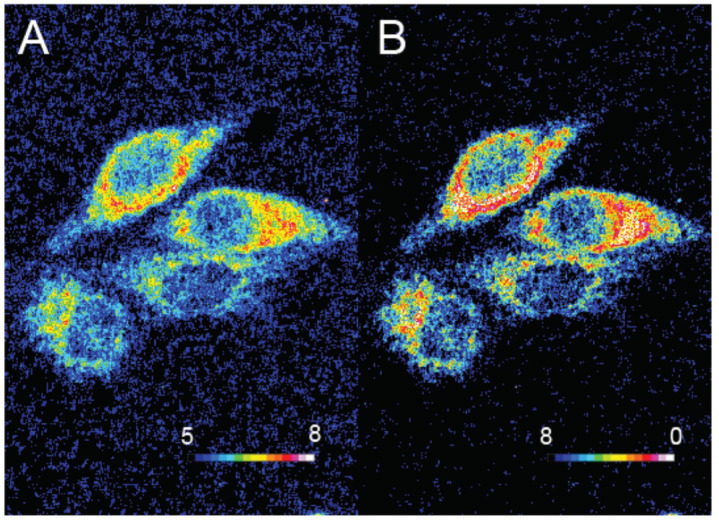
Ratiometric visualization of pH (**A**) and O_2_ concentration (mg/L) (**B**) in CaSki cells using described Pdots **16** (**C**). Reproduced with authorization from Xu [70].

**Figure 17 ijms-21-09217-f017:**
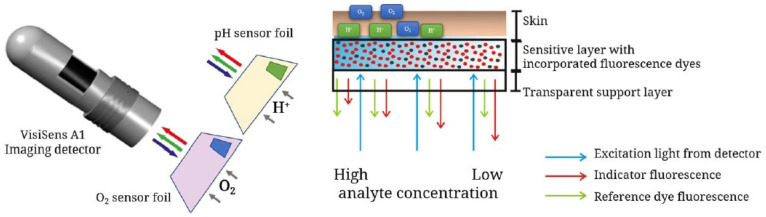
Recent schematic drawing of the pH and pO2 measurement of skin after radiation therapy from Auerswald [75].

**Figure 18 ijms-21-09217-f018:**
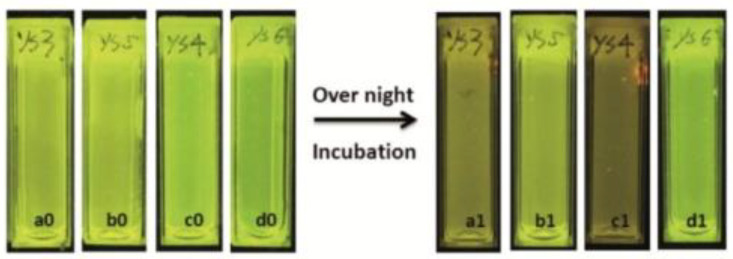
Fluorescence extinguishments due to the acidification of medium by the bacterial growth in a1 and c1. Reproduced with authorization from Si [83].

**Figure 19 ijms-21-09217-f019:**
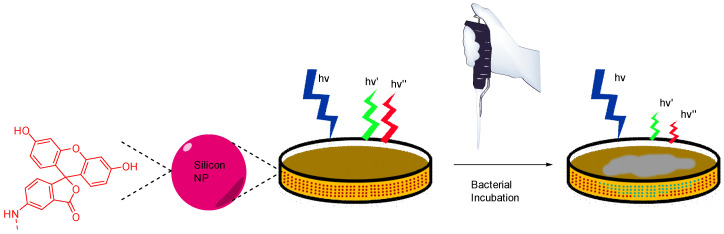
Bacterial and pH imaging using ratiometric signal from nanoparticles included in agar from Wang [84].

**Figure 20 ijms-21-09217-f020:**
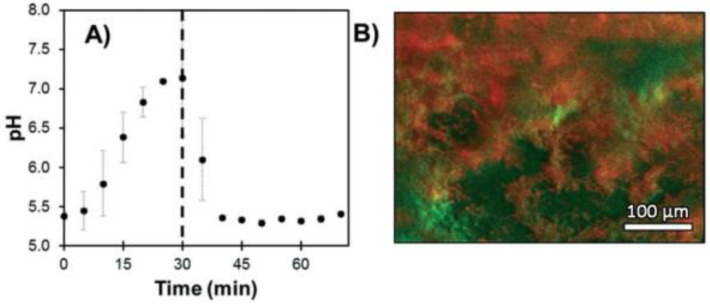
*S. salivarius *biofilm pH response under expositions to glucose solutions (**A**) and imaging of localized acidification (red) and biofilm accumulation (green) (**B**). Reproduced with authorization from Gashti [86].

**Figure 21 ijms-21-09217-f021:**
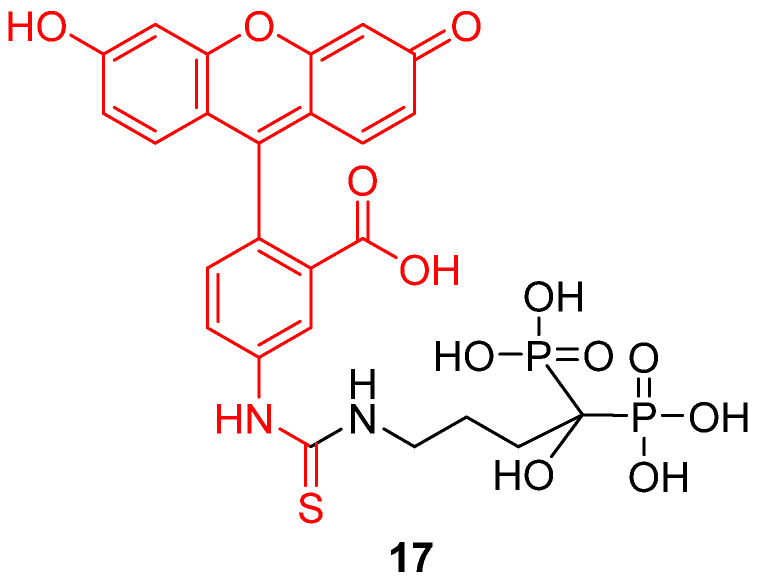
Bone pH-sensitive probe **17** according to Li [28].

**Figure 22 ijms-21-09217-f022:**
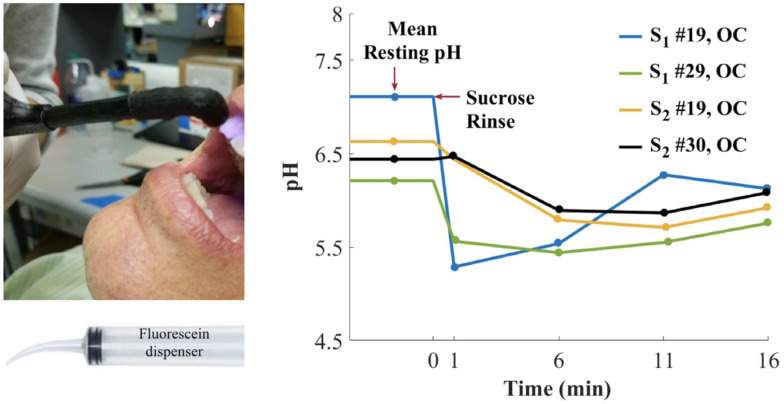
Dental pH monitoring using fluorescein fluorescence under exposition to glucose solutions, from Sharma [103].

**Figure 23 ijms-21-09217-f023:**
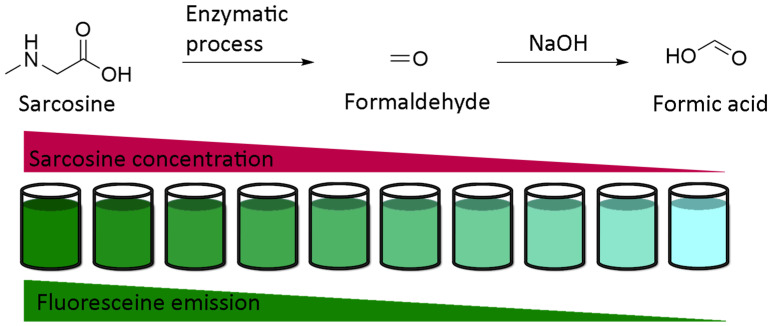
Titration of sarcosine using fluorescein according to Burton [104].

**Figure 24 ijms-21-09217-f024:**
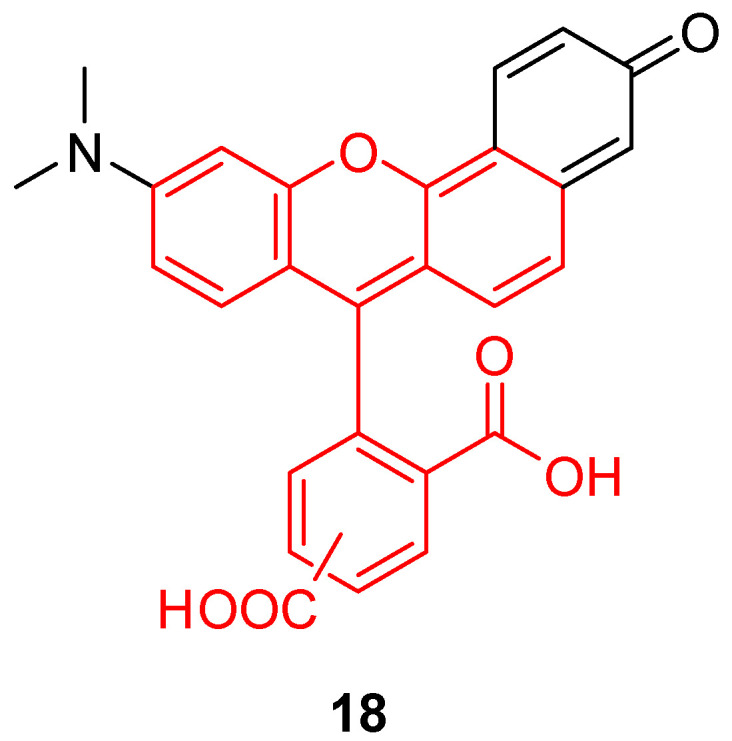
Chemical structure of Seminaphtharhodafluor, SNARF^®^
**18.**

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
