# Peer review of "Fluorescein Derivatives as Fluorescent Probes for pH Monitoring along Recent Biological Applications"

_ijms, 2020, doi:10.3390/ijms21239217_

Round 1
Reviewer 1 Report
Florent Le Guern an colleagues in their manuscript „Fluorescein derivatives as fluorescent probes for pH monitoring along recent biological applications“ provide a comprehensive review of the methods and applications that employ fluorescein derivatives for pH. The review describes applications that range from monitoring growth and metabolism of bacteria to measuring pH inside living eukaryotic cells. In my opinion, a brief overview of drawbacks and possible difficulties of using fluoresceins should be also included, as being aware of such information would greatly help the new users.
The review is broad and very well organized, and will be of interest to the community. Therefore, I am happy to recommend it for publication.
Author Response
Dear Reviewer,
Manuscript was modified and corrected according Reviewer’s suggestions. In addition to grammatical corrections, answers to suggestions are underlined in yellow in new manuscript. You will find in the following lines our answers to your different comments.
Comments and Suggestions for Authors
Florent Le Guern an colleagues in their manuscript „Fluorescein derivatives as fluorescent probes for pH monitoring along recent biological applications“ provide a comprehensive review of the methods and applications that employ fluorescein derivatives for pH. The review describes applications that range from monitoring growth and metabolism of bacteria to measuring pH inside living eukaryotic cells. In my opinion, a brief overview of drawbacks and possible difficulties of using fluoresceins should be also included, as being aware of such information would greatly help the new users.
ANSWER: We thank reviewer 1 for this suggestion. A section dealing with major fluorescein’s drawbacks was added, and starting from line 103 of the revised version.
Reviewer 2 Report
This review gives a comprehensive overview of the development of Fluorescein derivatives for pH sensing-related biological applications. Protons are involved in many physiological and pathological activities, while intracellular pH is important in consideration of normal cell activities. Abnormal intercellular pH and intracellular pH could reveal the presence of diseases including neurodegenerative and cardiopulmonary diseases. Due to the importance of pH monitoring in consideration of cell activities, metabolism and diseases, it is of importance to develop effective and sensitive pH probes. Contributed by its pH sensitivity and excellent fluorescent quantum yield, fluorescein has been a powerful tool as a pH-sensitive probe applied in pH-monitoring. This review describes the recent advances in the development of fluorescein derivatives, and their fluorescent pH-sensitive properties and applications in biological systems.
It would be better if this review could provide more detailed descriptions on the importance of pH monitoring in biology.
In Figure 2, the number of publications with keywords “fluorescein, pH, probe, biology” during last fifty years (around 13,000) is much higher than the other years. Any reason/comment for that?
Line 97: FITC is one of the most used derivatives due to its properties and good reactivity for conjugation. The conjugation linkage formed is a thiourea unit that may be less stable than the amide linkage formed by using FAM. The authors would consider to also comment on the stability of the bioconjugation linkage.
Line 110: since its main advantage was its low leaking rate through cell membranes due to its additional carboxylic acid function. Would the authors explain why the “additional carboxylic acid function” has the advantage of low leaking rate through cell membranes?
In Figure 14, the BSA is linked with the fluorescent dye and GA on the treated glass on a highly ordered manner. The conjugation of fluorescein on the BSA (probably through conjugation with lysine amino group) would not be site specific. Would the authors revise the representation of the Figure 14?
In Figure 15, the linkage of the sensible probes on the surface of AuNCs is through the SH group. In some literature, linkage through S- (thiolate anion) is reported. The authors are suggested to check the original literature to see whether SH or S- (thiolate anion) would be a better representation of the linkage.
The authors are suggested to proofread this review because grammatical mistakes are found in the context. Some prepositions are missing. The English writing could be better since some sentences sound strange and not academic.
Line 385: "In their view, Wang et al. were focused on pH monitoring.... ", "were" should be deleted.
Line 445: "The measurement of pH is also interesting in odontology", this sentence seems strange.
Please check the citation. e.g. In line 161, "Therefore, Hee Lee et al. put forward a novel dyad based on chemical association with rhodamine B", the surname of the first author is Lee, but not Hee Lee.
Author Response
Dear Reviewer,
Manuscript was modified and corrected according Reviewer’s suggestions. In addition to grammatical corrections, answers to suggestions are underlined in yellow in new manuscript. You will find in the following lines our answers to your different comments.
Comments and Suggestions for Authors
This review gives a comprehensive overview of the development of Fluorescein derivatives for pH sensing-related biological applications. Protons are involved in many physiological and pathological activities, while intracellular pH is important in consideration of normal cell activities. Abnormal intercellular pH and intracellular pH could reveal the presence of diseases including neurodegenerative and cardiopulmonary diseases. Due to the importance of pH monitoring in consideration of cell activities, metabolism and diseases, it is of importance to develop effective and sensitive pH probes. Contributed by its pH sensitivity and excellent fluorescent quantum yield, fluorescein has been a powerful tool as a pH-sensitive probe applied in pH-monitoring. This review describes the recent advances in the development of fluorescein derivatives, and their fluorescent pH-sensitive properties and applications in biological systems.
It would be better if this review could provide more detailed descriptions on the importance of pH monitoring in biology.
ANSWER: We agree with reviewer 2, additional details are of interest for a broad readership. The introduction was reworked, and importance of pH in biology is now highlighted from line 32 to 51.
In Figure 2, the number of publications with keywords “fluorescein, pH, probe, biology” during last fifty years (around 13,000) is much higher than the other years. Any reason/comment for that?
ANSWER: In fact, the unusual high number of hits in 2008 was due to publications of book chapters. The figure 2 (line 75) was modified by monitoring only published articles (without books chapters) which to our opinion is in agreement with the current publishing trend and now meaningful.
Line 97: FITC is one of the most used derivatives due to its properties and good reactivity for conjugation. The conjugation linkage formed is a thiourea unit that may be less stable than the amide linkage formed by using FAM. The authors would consider to also comment on the stability of the bioconjugation linkage.
ANSWER: After further investigations, the stability of thiourea adducts was studied. According to new ref 32-34, such adducts might suffer from instability depending on concentrations, pH and temperature. Comments are now included in lines 129-131 and 145-146 of the revised version.
Line 110: since its main advantage was its low leaking rate through cell membranes due to its additional carboxylic acid function. Would the authors explain why the “additional carboxylic acid function” has the advantage of low leaking rate through cell membranes?
ANSWER: The additional carboxylic acid moiety most likely modifies the anionic/neutral balance and the corresponding solubility in cell membranes. This information is now given in lines 141-143 of the revised version.
In Figure 14, the BSA is linked with the fluorescent dye and GA on the treated glass on a highly ordered manner. The conjugation of fluorescein on the BSA (probably through conjugation with lysine amino group) would not be site specific. Would the authors revise the representation of the Figure 14?
ANSWER: We agree with reviewer’s comment and modified figure 14 (line 299) accordingly, by cluttering BSA/FAM moieties.
In Figure 15, the linkage of the sensible probes on the surface of AuNCs is through the SH group. In some literature, linkage through S- (thiolate anion) is reported. The authors are suggested to check the original literature to see whether SH or S- (thiolate anion) would be a better representation of the linkage.
ANSWER: We agree with reviewer comment and modified figure 15 (line 320) accordingly, by representing this interaction with thiolate anions as reported in literature.
The authors are suggested to proofread this review because grammatical mistakes are found in the context. Some prepositions are missing. The English writing could be better since some sentences sound strange and not academic.
ANSWER: The whole manuscript was proofread once again by authors and an additional English-speaking colleague. A lot of grammatical mistakes were indeed found and corrected in this new version.
Line 385: "In their view, Wang et al. were focused on pH monitoring.... ", "were" should be deleted.
ANSWER: The sentence was corrected at line 394.
Line 445: "The measurement of pH is also interesting in odontology", this sentence seems strange.
ANSWER: The sentence was reworked at line 480.
Please check the citation. e.g. In line 161, "Therefore, Hee Lee et al. put forward a novel dyad based on chemical association with rhodamine B", the surname of the first author is Lee, but not Hee Lee.
ANSWER: The surname of the author has been modified at lines 194 and 204.
Reviewer 3 Report
The review is well written and edited.Topic is interesting and important and well described. The authors reviewed a number of publications on various fluorescein derivatives as fluorescent probes for pH monitoring. In general, I consider the article to be valuable and does not need to be corrected.
Author Response
Dear Reviewer,
Manuscript was modified and corrected according Reviewer’s suggestions. In addition to grammatical corrections, answers to suggestions are underlined in yellow in new manuscript.
Reviewer 3 : Comments and Suggestions for Authors
The review is well written and edited.Topic is interesting and important and well described. The authors reviewed a number of publications on various fluorescein derivatives as fluorescent probes for pH monitoring. In general, I consider the article to be valuable and does not need to be corrected.
ANSWER: Thanks you for your reviewing.